# Periostin Short Fragment with Exon 17 via Aberrant Alternative Splicing Is Required for Breast Cancer Growth and Metastasis

**DOI:** 10.3390/cells10040892

**Published:** 2021-04-14

**Authors:** Yuka Ikeda-Iwabu, Yoshiaki Taniyama, Naruto Katsuragi, Fumihiro Sanada, Nobutaka Koibuchi, Kana Shibata, Kenzo Shimazu, Hiromi Rakugi, Ryuichi Morishita

**Affiliations:** 1Department of Clinical Gene Therapy, Osaka University Graduate School of Medicine, Suita, Osaka 565-0871, Japan; kirarytypopy@yahoo.co.jp (Y.I.-I.); katsuragi@cgt.med.osaka-u.ac.jp (N.K.); sanada@cgt.med.osaka-u.ac.jp (F.S.); koibuchi@cgt.med.osaka-u.ac.jp (N.K.); shibata@periotherapia.co.jp (K.S.); 2Department of Geriatric and General Medicine, Osaka University Graduate School of Medicine, Suita, Osaka 565-0871, Japan; rakugi@geriat.med.osaka-u.ac.jp; 3Department of Breast and Endocrine Surgery, Osaka University Graduate School of Medicine, Suita, Osaka 565-0871, Japan; kenzoshimazu@gmail.com

**Keywords:** periostin, splicing variants, triple negative cancer, wnt3a

## Abstract

Background: Periostin (POSTN) is a 93 kDa matrix protein that helps to regulate collagen gene expression in the extracellular matrix. POSTN overexpression is a prognostic factor in malignant cancers; however, some researchers have observed it in the stroma, whereas others have reported it on tumors. Objective: This study aimed to investigate the function of POSTN on tumors. Methods and Results: We found that POSTN in cancer cells can be detected by using an antibody against the POSTN C-terminal region exon 17 (Ex17 antibody), but not with an antibody against the POSTN N-terminal region exon 12 (Ex12 antibody) in patients with breast cancer. In a fraction secreted from fibroblasts, LC–MS/MS analysis revealed a short fragment of POSTN of approximately 40 kDa with exon 17. In addition, molecular interaction analysis showed that POSTN with exon 17, but not POSTN without exon 17, bound specifically to wnt3a, and the Ex17 antibody inhibited the binding. Conclusion: A short fragment of POSTN with exon 17, which originates in the fibroblasts, is transported to cancer cells, whereas POSTN fragments without exon 17 are retained in the stroma. The Ex17 antibody inhibits the binding between POSTN exon 17 and wnt3a.

## 1. Introduction

Periostin (POSTN) is an extracellular matrix (ECM) protein that is highly expressed in several human cancers, including breast cancer. Clinical studies on POSTN expression have shown that elevated levels of serum POSTN or tissue POSTN are associated with the increased malignant behavior of multiple types of cancer, such as melanomas [1], lung metastases [2], and colon [3], pancreatic [4], esophageal [5], and ovarian [6] cancers. Moreover, in animal models, POSTN suppression was reported to improve the progression of cancer variants [7,8]. Triple-negative breast cancer (TNBC) is a subtype with no targeted therapy because patients with TNBC are negative for the estrogen, progesterone, and HER2 receptors. TNBC occurs in approximately 10–15% of patients with breast cancer [9]. It is unclear why several reports have indicated that POSTN is highly expressed in metastatic cancers [1,2,3,4,5,6]. However, it has been reported that inhibiting the ability to produce an ECM protein via ECM-producing stromal cells in the local microenvironment of MDA-MB-231 cancer cells induces alterations in their metastatic tropism, suggesting that there may be various forms of ECM proteins as per their different functions [8,10].

The identification and characterization of POSTN initially started with restricted expression to the periosteum and periodontal ligament and increased expression by transforming growth factor-β (TGF-β) [11]. POSTN shares a homology with the insect cell adhesion molecule fasciclin I (FASI) and the human βIgH3 (TGFBI) protein, both of which are induced by transforming growth factor-β (TGF-β) [8,12]. Considering the above, POSTN has a similar domain structure to several secreted ECM proteins, such as fibronectin, tenascin-C, and osteopontin; hence, it may be available for action in the stromal cells [13,14,15]. In addition, a previous report noted that, in ECM remodeling, POSTN can be effectively induced via various signaling pathways, such as with bone morphogenic protein-2 (BMP 2) to promote inflammation and develop metastatic breast cancer MDA-MB-231 [16], or with connective tissue growth factor 2, angiotensin II, and mechanical stress. In allergic skin inflammation, interleukin (IL)-4 and IL-13 derived from Th2 cells stimulate fibroblasts to produce POSTN, which binds to αv integrin, inducing proinflammatory cytokines [17].

Each ECM protein can be structurally spliced into variant forms, and POSTN also has variant C-terminal splicing forms [12]. Therefore, whether or not full-length POSTN can exist in human breast tumor cells remains controversial. We hypothesized that the localization of a POSTN fragment could vary in cancer–stroma interactions if there was a POSTN fragment protein with exon 17 in the POSTN C-terminus region, but not if there was exon 12 in the POSTN N-terminus region.

In animal models, POSTN overexpression in the stroma is associated with metastasis, angiogenesis, and epithelial–mesenchymal transition via integrin αvβ3 and αvβ5 binding [18]. Malanchi et al. reported that POSTN expressed by fibroblasts in primary tumor cells is required for cancer stem cell maintenance, and may act as a critical tumorigenesis and progression regulator [7]. POSTN is primarily localized in stromal cells [19]; however, Xu et al. reported that POSTN may exist near and around cancer cells. Furthermore, the possibility of POSTN as a potential biomarker of the metastasis and chemotherapy resistance of breast cancer in a study of 1086 patients with breast cancer was noted [20]. Using clinicopathological data from patients with breast cancer, Ratajczak-Wielgomas et al. recently reported that POSTN expression is localized in both cancer-associated fibroblasts and tumor cells, which suggests that POSTN may act as a critical regulator at various stages of tumorigenesis and progression [21]. In another report, POSTN protein expression was increased in CD44^high^/CD24^low^ breast cancer stem cells, compared with control cells, and associated with cancer stem cell chemotherapy resistance [22]. The abovementioned studies demonstrated that POSTN may be critical for interactions between cancer stem cells and their metastatic niche [7,8]. Therefore, although the functional differences among POSTN in the stroma and cancer cells in many cancer types remain unclear, it is evident that both variants of POSTN are poor prognostic markers of breast cancer.

To clarify the above differences, we focused on different POSTN splicing variants (PN1–4). We previously reported the importance of full-length POSTN (PN1) in a 4T1 mouse TNBC model, and we showed that PN1 inhibition by a polyclonal antibody against exon 17 of the rat POSTN C-terminus decreased primary tumor size and inhibited lung metastasis [23]. The clinical significance of the POSTN C-terminal region is poorly understood. We hypothesized that the POSTN C-terminus plays an important role in breast cancer progression and metastasis. In general, POSTN in the stroma is identified using antibodies against the POSTN N-terminal. However, we report in this study that the effects of POSTN on cancer can be detected by antibodies against the POSTN C-terminal, which suggests that different POSTN variants may exist in different regions in the cancer microenvironment.

POSTN has four splicing variants: PN1 (full-length POSTN), PN2 (absent exon 17), PN3 (absent exon 21), and PN4 (absent exons 17 and 21). We focused on PN1 and PN3, which contain POSTN exon 17. We have reported on the importance of POSTN exon 17 in breast cancer [23], heart failure [24,25,26], cerebral infarction [27,28], diabetic retinopathy [15], and knee osteoarthritis [14] in animal models; however, the function of POSTN exon 17 in pathological organs such as the primary and metastatic regions of breast cancer is still unclear.

In this study, we compared the distribution of POSTN variants in breast cancer using antibodies against POSTN exon 17 (C-terminal) and POSTN exon 12 (N-terminal) and investigated the association between POSTN exon 17 and wnt3a.

## 2. Materials and Methods

### 2.1. Animals

Female NOG mice (NOD/Shi-scid, IL-2RγKO) were obtained from Charles River Laboratories Japan (Tokyo, Japan). They were bred and maintained in the animal facility at Osaka University as per institutional guidelines. All animal experiments were performed in accordance with the protocols approved by the Animal Ethics Committee of Osaka University. We selected MDA-MB-231 human TNBC cell-line variants in vivo for their aggressive spontaneous metastatic properties after the mice were inoculated with TNBC. These were cultured in DMEM medium with 10% fetal bovine serum (FBS) at 37 °C in 5% CO_2_, as previously described [23]. Surgical procedures with the MDA-MB-231 metastatic variants were performed as previously described [23]. We implanted 2 × 10^6^ cells from the MDA-MB-231 cell line in the mammary fat pads of 8-week-old female NOG mice.

### 2.2. Cell Lines and Reagents

We obtained human breast cancer (MDA-MB-231) cells from the American Type Culture Collection (Manassas, VA, USA). The cells were maintained in DMEM (Nacalai Tesque, Kyoto, Japan), supplemented with 10% (vol/vol) fetal bovine serum (FBS) (Biowest, Nuaille, France) and 1% penicillin/streptomycin (Nacalai Tesque, Kyoto, Japan) under 5% CO_2_ at 37 °C. We primarily isolated fibroblasts from the lungs of C57BL/6N wild-type (WT) mice, POSTN exon 17 knock out (KO) mice, and POSTN-null mice [15].

### 2.3. Antibodies against POSTN

We generated the monoclonal Ex17 antibody in immunized mice using human POSTN exon 17 peptides with keyhole limpet hemocyanin (KLH), as previously described [23,25]. We tested the specificity of this antibody with epitope mapping, as shown in Figure 1H,I. Ex12 antibody (AG-20B-0033-C100, Adipogen, San Diego, CA, USA) [7] recognizes POSTN exon 12, and Ex17 antibody [25] recognizes POSTN exon 17. The two main antibodies against POSTN exon 12 or exon 17 were used for all analyses in this paper.

### 2.4. Dot Blotting

Dilutions of 1 mg/mL (stocked concentration 1 mg/mL) of human POSTN exon 1–23 peptides were prepared in phosphate-buffered saline dilution buffer. Each POSTN exon peptide was separately spotted and allowed to dry in 2–5 mL volumes on the nitrocellulose membrane (0.2 μm). Epitope mapping with POSTN antibody was a simple way to identify which peptide sequence of the POSTN exon bound to each sample of POSTN antibody. Dot blotting using Western blotting easily identified the antigen of POSTN antibodies.

### 2.5. Immunohistological Assay

For the immunohistochemistry of human breast cancer patients, we received tumor samples in optical coherence tomography (OCT) from the laboratory of the Department of Breast and Endocrine Surgery at Osaka University. We made frozen tissue sections without antigen retrieval because we tried to identify the POSTN variant present in cancers using a proteome analysis of frozen tissue section slides. We used glass slides with foils (Leica, Wetzlar, German), and each frozen section with a section thickness of 10 μm was attached to a glass slide. The staining was detected using a polymer-based detection method, and we used POSTN Ex17 antibody or POSTN Ex12 antibody as the primary monoclonal antibodies. The secondary antibody comprised of horseradish peroxidase (HRP)-labeled polymer reagents for human tissues against each primary antibody (MAX-PO (MULTI), Code: 724152, Nichirei, Tokyo, Japan). The HRP-3,3′-Diaminobenzidine (DAB) method was used for the chromogenic detection of immunostaining. The stained images of each sample were taken using an Olympus BX51 digital camera DP71.

For the tissue immunofluorescence staining, tumor sections from the mice were embedded in OCT compound (Sakura, Tokyo, Japan), and frozen tissue sections were prepared. The sections were fixed with 0.3% hydrogen peroxide solution/methanol and stained with antibodies against POSTN (AG-20B-0033-C100, Adipogen, San Diego, CA, USA) the epitope of which was determined to be exon 12 in this paper, or a neutralizing antibody against POSTN exon 17 (monoclonal antibody, POSTN Ex17 antibody and PN1-antibody are identical) [23]. The stained images of each sample were taken using a Keyence BZ-X800 (Osaka, Japan).

For immunocytochemistry, MDA-MB-231 cells were cultured in DMEM supplemented with 5% (vol/vol) FBS and 1 % penicillin/streptomycin. After washing, the cells were incubated for 5 min at ambient temperature in 100 % methanol, then cooled to −20 °C for cell fixation. We stained the MDA-MB-231 cells using the Ex17 antibody, conjugated with Alexa Fluor 488 (green). In addition, we stained the cell organelles without transparency processing and used DAPI for nuclei staining (blue). The stained images of each sample were captured using a Keyence BZ-X800.

### 2.6. Western Blotting

For the immunoblotting analysis, the cells (1 × 10^6^ cells/mL) were treated with human TGF-β3 (10 ng/mL) (Recombinant Human TGF-β3, 243-B3-002, R&D Systems, Minneapolis, MN, USA) for 12 h, and then lysed in Cell Lysis Buffer (CLB) (ab152163, Abcam, Cambridge, UK; 0.216% beta glycerophosphate, 0.19% sodium orthovanadate, 0.001% leupeptin, 0.38% EGT, 10% Triton-X-100, 3.15% Tris HCl, 8.8% sodium chloride, 0.29% sodium ethylenediaminetetraacetic acid (EDTA), 1.12% sodium pyrophosphate decahydrate) with protease inhibitor mixture (Nacalai Tesque, Kyoto, Japan) on ice, and centrifuged at 14,000× *g* for 20 min at 4 °C. Then, the supernatants from the fibroblasts were prepared using a protein concentration kit (Amicom Ultra-15 NMWL 3000, Millipore, Burlington, MA, USA) and the protein extracts were quantified using a protein quantification assay kit (Reagent A Cat.#500-0113, Reagent S Cat.#500-0115, Bio-Rad, Hercules, CA, USA). The protein extracts in the Laemmli sample buffer (Cat. #161-0747, Bio-Rad, Hercules, CA, USA) were resolved by SDS-PAGE and transferred onto Hypond-P membranes (GE Healthcare, Chicago, IL, USA). The membranes were immunoblotted with the indicated antibodies, and the bound antibodies were visualized with horseradish peroxidase-conjugated antibodies against rabbit or mouse IgG using the ECL Western blotting system (1:10,000, GE Healthcare, Chicago, IL, USA). Primary antibodies against POSTN exon 17, POSTN exon 12 (AG-20B-0033-C100, Adipogen, San Diego, CA, USA), and β-actin (#4967, 1:1000, Cell Signaling, Danvers, MA, USA) were used. The Ex17 antibody were generated in immunized mice and purified using an affinity column [23]. The band intensities were quantified using ImageJ software (National Institutes of Health (NIH), Bethesda, Maryland USA).

### 2.7. Immunoprecipitation

Whole-cell extraction and the supernatant from MDA-MB-231 or fibroblast cells were lysed using CLB lysis buffer and immunoprecipitated with the Ex17 antibody or normal immunoglobulin G antibodies. Protein A/G dynabeads (1003D, ThermoFisher, Waltham, MA, USA) were applied, and the last-precipitated proteins were subjected to immunoblotting with the Ex17 antibody.

### 2.8. Liquid Chromatography–Tandem Mass Spectrometry (LC–MS/MS) and Data Analysis

Lung fibroblasts were isolated from the pups of POSTN exon 17 KO mice and null mice. These mice have been previously described, and were backcrossed at least seven times with C57BL/6N WT mice (CLEA, Tokyo, Japan). To characterize the POSTN exon 17 KO mice, littermate WT (C57BL/6N) and null mice, which were phenotypically indistinguishable, were used as controls. Polymerase chain reactions (PCR) was used to determine the genotypes of the experimental mice. According to the method previously described [29], LC–MS was performed using an LC– MS/MS system equipped with two LC-30 AD pumps, a DGU-20As degasser, an SIL-30 AC autosampler, a CTO-20 AC column oven, and a CBM-20A control module, coupled with an LCMS-8040 triple quadrupole mass spectrometer (Shimazu Co., Kyoto, Japan). Target bands matching the POSTN Ex17 antibody-cross-reacting bands were excised from the gel or the membrane and analyzed using matrix-assisted laser desorption/ionization-quadrupole-time-of-flight-tandem mass spectrometry (MALDI- quadrupole time-of flight (QqTOF) MS/MS).

### 2.9. Surface Plasmon Resonance Analysis

For the protein binding analysis, biotinylated recombinant human PN1 protein (3548-F2-050, R&D Systems, Minneapolis, MN, USA) or mouse PN2 protein (2955-F2-050, R&D Systems, Minneapolis, MN, USA) was immobilized on an SA sensor chip (BR-1005-31, GE Healthcare) by 0.1 μM, and the analysis was performed by injecting 100 nM of human wnt3a (W3a-H-005, StemRD 3H11104) or BSA over the surface of the chip in the Biacore T200 system (GE Healthcare, Chicago, IL, USA). Biotinylated PN variants were prepared using a Biotin CAPture Kit (Series S 28920234, GE Healthcare). The final immobilization levels were approximately 2000 response units. These experiments were performed at 4 °C at a constant flow rate of 30 µL/min of driving buffer. We used the running buffer (4-(2-hydroxyethyl)-1-piperazineethanesulfonic acid (HEPES), 0.05% Tween 20) as the driving buffer, and all buffers were filtered before use and degassed.

For the kinetic analysis, the recombinant human PN1 protein or mouse PN2 protein was immobilized on a sensor chip CM5 (BR-1005-30, GE Healthcare, Chicago, IL, USA) at pH 5.0 by 0.1 μM, and the fitting curve analysis was performed by injecting 100 nM of human wnt3a or BSA over the surface of the chip in the Biacore T200 system (GE Healthcare, Chicago, IL, USA). The CM5 sensor chip was activated with EDC/NHS for amine coupling (Amin Coupling Kit, Code:BR-1000-50, GE Healthcare, Chicago, IL, USA). For the single-cycle kinetics, we ran a series of analyte concentrations in one cycle for kinetic evaluation. These experiments were performed at 4 °C at a constant flow rate of 30 µL/min of driving buffer.

For the inhibition experiment using Ex17 antibody, 20 μg/mL of Ex17 antibody was immobilized on a CM5 sensor chip (BR-1005-30, GE Healthcare, Chicago, IL, USA) at pH 5.0. One program was made as follows: STEP (1) to STEP (4) with each step repeated twice. STEP (1) 10 nM PN1 protein was injected. STEP (2) A mixture of 10 nM PN1 protein and 100 nM wmt3a was injected after washing at pH 3. STEP (3) 100 nM Wmt3a was injected after washing at pH 3. STEP (4) 10 nM PN1 protein was injected after washing at pH 3. These experiments were performed at 4 °C at a constant flow rate of 10 µL/min of driving buffer. The contact time was maintained at 1 min in all the analyses.

### 2.10. Total RNA Extraction via Quantitative Real-Time PCR

Total RNA was isolated from the MDA-MB 231 cells and fibroblasts using an RNeasy kit (Qiagen, Hilden, Germany) and was reverse-transcribed with Super Script III First-Strand Synthesis SuperMix (Invitrogen). The expression levels of the POSTN exon 17, POSTN exon 21, and Axin2 target genes were determined by quantitative real-time PCR and normalized to the level of Glyceraldehyde 3-phosphate dehydrogenase (GAPDH). The following primer sets were used for the mouse POSTN variant analysis: mouse PN1-forward, 5′-ATAACCAAAGTCGTGGAACC-3′; mouse PN1-reverse, 5′-TGTCTCCCTGAAGCAGTCTT-3′; mouse PN2-forward, 5′-CCATGACTGTCTATAGACCTG-3′; mouse PN2-reverse, 5′-TGTCTCCCTGAAGCAGTCTT-3′; mouse PN3-forward, 5′-ATAACCAAAGTCGTGGAACC-3′; mouse PN3-reverse, 5′-TTTGCAGGTGTGTCTTTTTG-3′; mouse PN4-forward, 5′-CCCCATGACTGTCTATAGACC-3′, mouse PN4-reverse, 5′-TTCTTTGCAGGTGTGTCTTTT-3′.

The following primers were used for the wnt3a downstream analysis: Axin2-forward, 5′-CTGGCTCCAGAAGATCACAAAG-3′, Axin2-reverse, 5′-CATCCTCCCAGATCTCCTCAAA-3′; GAPDH-forward, 5′-CAAGCTCATTTCCTGGTATGACAAT-3′; GAPDH-reverse, 5′-GTTGGGATAGGGCCTCTCTTG-3′.

### 2.11. Statistical Analysis

The statistical analysis results are shown as the means ± SD. The Mann–Whitney test was performed for comparing multiple treatment groups. For the statistical analysis of the expression change of genes from two groups, the Wilcoxon signed-rank test was performed.

## 3. Results

### 3.1. POSTN Exon 17 Is Associated with Breast Cancer Cells

To examine the localization of POSTN according to the epitope of the POSTN antibody, we used primary tissue obtained from patients with breast cancer. Immunohistochemical examination showed that the Ex17 antibody was bound to cancer cells on the breast duct epithelium. (Figure 1A–E) In another patient, breast cancer cells stained positive for the Ex17 antibody, although the stromal cells were slightly stained. In contrast, sections from patients with breast cancer were completely stained in the stroma, but the cancer cells were only slightly stained using a commercially available POSTN monoclonal antibody (Ex12) (Figure 1A–E). We confirmed its epitope, POSTN exon 12, using a dot blot (Figure 1H,I).

Negative staining was performed under reaction conditions, excluding the primary antibody reaction. These were nonspecific reactions because the samples were frozen sections from patients with breast cancer. However, the staining pattern using the Ex17 antibody significantly differed from the pattern obtained using the Ex12 antibody when we examined five patients with breast cancer (Figure 1A–E).

To investigate POSTN isoforms in the mouse breast cancer model, we attempted to create animal metastatic models using the human breast cancer cell line, MDA-MB 231. In NOG mice transplanted with human TNBC (MDA-MB 231 cell line), the immunohistochemical analysis of subcutaneous tissue of the primary and metastatic lesions demonstrated that POSTN expression differed by location, depending on whether the POSTN isoforms contained exon 17 in the TNBC cells. (Figure 1F,G) This result was similar with the immunostaining pattern in cancer cells from patients with breast cancer, and the staining pattern using the Ex17 antibody was observed in only a part of the cancer tissue. To confirm the relationship between high levels of POSTN and the isoforms, we used the Ex17 and Ex12 antibody for immunostaining, and evaluated the epitope (Figure 1H,I). In the primary and metastatic lesions in the lung, POSTN exon 12 localized in the stroma (Figure 1F). This result was similar to the immunostaining pattern observed in cancer cells from patients with breast cancer.

### 3.2. Proteomic Approach for the Identification of Short Fragments of POSTN

When we performed Western blotting using Ex17 antibody after immunoprecipitation with Ex17 antibody, we detected two bands of approximately 40 kDa in the cell lysate and the supernatant of lung fibroblasts with-or-without TGF-β3 stimulation (Figure 2(A1,2)). We analyzed this short fragment using LC–MS/MS and found that POSTN exon 18 was present in it (Figure 3A). However, we detected a band of approximately 40 kDa in the cell lysate and supernatant of MDA-MB-231 with-or-without TGF-β3 stimulation (Figure 2(B1,2)).

In addition, when we performed Western blotting using Ex12 antibody, we detected a band of about 75 kDa in the cell lysate and supernatant of lung fibroblasts with-or-without TGF-β3 stimulation (Figure 2(A1,2)). We analyzed this fragment using LC–MS/MS and found POSTN exons 7, 10, 14, and 15 within it (Figure 3B). However, we did not detect any band of about 75 kDa in the cell lysate and supernatant of MDA-MB-231 with-or-without TGF-β3 stimulation.

### 3.3. POSTN Exon 17 Region Binds to Wnt3a

We tested whether POSTN with exon 17 could directly bind to wnt3a (Figure 4). PN1 (with exon 17) and PN2 (without exon 17) were immobilized after biotinylation at sensor chip streptavidin (SA), and bovine serum albumin (BSA) as a control, or wnt3a were flowed into the system. After washing, wnt3a bound tightly to PN1 instead of to PN2. (Figure 4A,B) The full-binding kinetic analysis of the POSTN Ex17 antibody was performed using Biacore 3000. Detailed binding kinetic parameters (association rate, ka; dissociation rate, kd; affinity constant, KD) can be determined via a full kinetic analysis (Figure 4C–E).

According to our data, the immobilized recombinant POSTN protein with exon 17 bound with the Ex17 antibody, and the association between wnt3a and POSTN with exon 17 was inhibited by the POSTN Ex17 antibody (Figure 4F,G). However, the binding between wnt3a and POSTN without exon 17 was weak because the bond was easily resolved after washing.

## 4. Discussion

In this study, we found that the POSTN short fragment containing exon 17 but not exon 12 and primarily originates from fibroblasts, was cleaved from its N-terminal region and accumulated in the cancer cell ECM, while the POSTN fragment containing exon 12 remained in the stroma. We report that the short fragment of POSTN containing exon 17, induced tumor invasion and lung metastasis because the Ex17 antibody treatment inhibited these actions. POSTN exon 17 may acquire new functions via cleavage by proteases because proteins usually change their function by simple fragmentation.

POSTN, a secreted matricellular N-glycoprotein of 93 kDa, was originally identified in the mouse osteoblast cell line NIH3T3 [11] and showed effects on bone regeneration. The POSTN N-terminal region (exons 1–15), which includes the FASI domains, is conserved in a variety of species; however, the C-terminal region (exons 16–23) gives rise to different splice isoforms by alternative splicing (mainly PN1–4) (Figure 1G) The POSTN region, between exon 17 and 21, is subject to widespread alternative splicing [12].

POSTN was previously identified in stromal cells surrounding a variety of malignant tumors [19]. In gene expression profiling, POSTN is expressed at the mRNA level in cancer-associated fibroblasts. Increased POSTN expression in the stroma surrounding primary and metastatic tumors or in the serum of patients with malignant tumors is considered a negative prognostic biomarker [19,20,21,30]. However, in 2004, Shao et al. [30] demonstrated that POSTN was observed in cancer cells evaluated by immunohistochemistry; thus, it would be transported through increased blood vessels to the surrounding cancer cells.

The differing roles of POSTN expression in stromal or cancer cells remain largely unknown. In this study, we examined and confirmed its therapeutic effect against mouse models of TNBC using the monoclonal Ex17 antibody. Furthermore, to identify the roles of the POSTN C-terminal region in tumorigenesis, we performed immunostaining on sections isolated from patients with breast cancer using the monoclonal Ex17 antibody.

Thus far, more than half of patients with breast cancer with lymph node metastasis have displayed the induction of stromal POSTN expression [7,20]. Another study reported that POSTN was located in the cytoplasm and membrane of cancer cells from patients with breast cancer with lymph node metastasis [20,21]. Our results showed that POSTN was highly expressed in patients with breast cancer or mouse tumor models when immunostaining was performed on the sections that were isolated using the monoclonal Ex17 antibody. In such cases, and localization occurred in cancer cells rather than in the stromal region.

Malanchi et al. reported that 75% of patients with breast cancer with lymph node metastases showed the induction of stromal POSTN by immunohistochemical staining with a commercially available monoclonal antibody [7]. We performed epitope mapping of this anti-POSTN antibody by dot blot (Figure 1H,I). This antibody detected the POSTN N-terminal exon 12 region and the exon 12 antibody stained the stroma surrounding the tumor cells, but not the cancer cells. Contrary to the results of the Ex12 antibody, the Ex17 antibody [23,25] stained the tumor cells but not the stroma cells, in vivo.

To analyze this phenomenon in detail, we detected the low molecular weight POSTN protein fragment that contained exon 17 from the supernatant of fibroblasts and MBA-MB231 breast cancer cells by Western blotting with the Ex17 antibody, and we used it for the following in vitro experiment. We detected two bands of approximately 40 kDa by mass spectrometry and recognized the signal involving parts of the POSTN exon 18 region in this short fragment. Taken together, short fragments of approximately 40 kDa in size obtained from fibroblast supernatant fractions, are parts of POSTN that contain exon 17 but not exon 12. The short fragments may be parts of PN1 or PN3, which both contain POSTN exon 17 (Figure 1G).

We detected bands of approximately 75 kDa from the fibroblast supernatant fractions by Western blotting with the Ex12 antibody and performed silver staining after SDS-PAGE for mass spectrometry using the same fraction. In the 75 kDa bands, several sequences included three types of signal. Parts of POSTN exon 7, exon 10, and the second half of exon 14–exon 15, were contained and easy to detect by mass spectrometry (Figure 3B). Taken together, the 75 kDa POSTN fragments represent the part of POSTN that contains exon 12.

A short fragment of POSTN with exon 17 at about 40 kDa (Figure 2(A1,2),(B1,2)) may be parts of PN1, PN3, or both, and a long fragment of POSTN with exon 12 at about 75 kDa (Figure 2(A3,4),(B3,4)) may be parts of PN2, PN4, or both. We cannot clarify these variants, so further study is needed in future. As a result, the secreted POSTN short fragment of approximately 40 kDa which contains exon 17 but not exon 12, may be transported to cancer cells, whereas the POSTN fragment of approximately 75 kDa which contains exon 12 may remain in the stroma.

The short fragment of POSTN containing exon 17 which originates from fibroblasts, may be transported to cancer cells, but significant questions remain. Why are we unable to detect the POSTN short fragment containing exon 17 from fibroblasts in the stroma even though there are many fibroblasts in the stroma? Why can we detect POSTN exon 17 on both the surface and interior of cancer cells even though the POSTN short fragment accumulates mainly on the surfaces of cancer cells? Vardaki et al. reported that POSTN is transported by exosomes from MDA-MB-231 human TNBC or 4T1 mouse TNBC cells [31], and Luga et al. reported that fibroblasts in the stroma promote breast cancer cell protrusive activity [32]. There is cell–cell communication between breast cancer cells and fibroblasts which occurs via exosomes. If the POSTN short fragment with exon 17 from the fibroblasts exists inside the exosome, we cannot detect POSTN exon 17 by immunohistochemistry in the stroma. It is possible that the exosome is transported to cancer cells, captured inside, and then broken down (Figure 5). Finally, we can detect the POSTN short fragment containing exon 17 inside cancer cells without exosomes by immunohistochemistry. Moreover, we may be able to detect the POSTN short fragment with exon 17 in exosomes because lysis buffer was used to extract the membrane proteins for POSTN protein analysis in this study (Figure 3).

Breast cancer produces a short fragment of POSTN (Figure 2(B1,2)), so, at least in part, the Ex17 antibody would detect it inside the cancer cell. The secreted POSTN protein’s N-terminal region with exons 1–15 is composed of a signal sequence, a cysteine-rich EMILIN-1 (EMI) domain, and four homologous fasciclin 1 (FAS1) domains. Although the POSTN region from exon 16 to exon 23 is a unique C-terminal region, POSTN paralog human βIgH3 (TGFBI) is a protein with a domain structure identical to the human POSTN N-terminal region, which spans from exon 1 to exon 15 [12].

How was the short fragment of POSTN with exon 17 produced? It was reported that the cleavage of the FAS1 domain occurs via proteolytic degradation by a single mutation of the FAS1 domain of the βIgH3 (TGFBI) protein with human granular corneal dystrophy type 1 [33]. Moreover, the proteolytic cleavage of the POSTN C-terminal end has been suggested to be essential for ECM protein binding or ECM structural stability [34]. The short fragment of POSTN with exon 17 may be produced because the cleavage of the FAS1 domain occurs via proteolytic degradation.

Despite the report of the predicted molecular weight of POSTN, which is approximately 90 kDa, a molecular weight of approximately 40 kDa was confirmed by our Ex17 antibody from the bands observed from the supernatant or cell lysate fraction of fibroblasts or cancer cells. POSTN exon 14 undergoes translational modification with N-glycosylation in the Golgi. Hence, the POSTN fragment with exon 17 would also be transported via the Golgi and may be modified by proteasomes.

It is likely that ECM proteins such as POSTN with several variant forms may have different membrane vesicle trafficking pathways according to each variant, because there are different translational modification types for each variant sequence [31,35]. Therefore, we can hypothesize that POSTN short fragments are located inside the exosomes.

During the process of post-translational modification, POSTN might be cleaved by many proteases in the Golgi or by extracellular proteases such as matrix metalloproteinases (MMPs) near the cell surface. Therefore, PN1 and PN3, which include exon 17, may be cleaved to diversify the protein’s function, and the low molecular weight POSTN protein fragment which includes exon 17, accumulates in the ECM of cancer cells.

Malanich et al. conversely reported that, when POSTN combines with wnt3a in the breast cancer cell, it activates wnt signaling [7]. We compared the binding between POSTN variants and wnt3a [36]. Surface plasmon analysis confirmed that PN-1 with exon 17 was strongly bound to wnt3a but PN-2, which lacks exon 17, was not. Wnta3a was reported to bind to chondroitin sulfate proteoglycans (one of the matrix proteins) on the cell surface [37]. POSTN, a matrix protein, also eFxists in the proteoglycan layer on the cell surface. The combination between wnt3a and POSTN appears to be natural.

While POSTN in the stroma was stained by the Ex12 antibody, but not the Ex17 antibody, the stromal fibroblasts secreted POSTN variants without exon 17 (PN2 or 4). These POSTN variants do not bind tightly to wnt3a, and would thus have different functions. Based on our results, the difference in each POSTN short variant depends on the binding strength to wnt3a.

In conclusion, we found that a short fragment of POSTN from fibroblasts and breast cancer cells containing exon 17 but not exon 12, binds tightly to wnt3a. The Ex17 antibody may prevent the binding between POSTN and wnt3a. Thus, our results highlight a new candidate for breast cancer treatment.

## 5. Conclusions

In conclusion, we found that a short fragment of POSTN that contains exon 17 but not exon 12 strongly binds wnt3a, and Ex17 antibody may prevent the growth of primary and metastatic breast cancer. Thus, our results highlight a new candidate for breast cancer treatment.

## 6. Patents

The patent of Ex17 antinody belongs to Osaka University, Periotherapia Co., which has the priority negotiation right.

## Figures and Tables

**Figure 1 cells-10-00892-f001:**
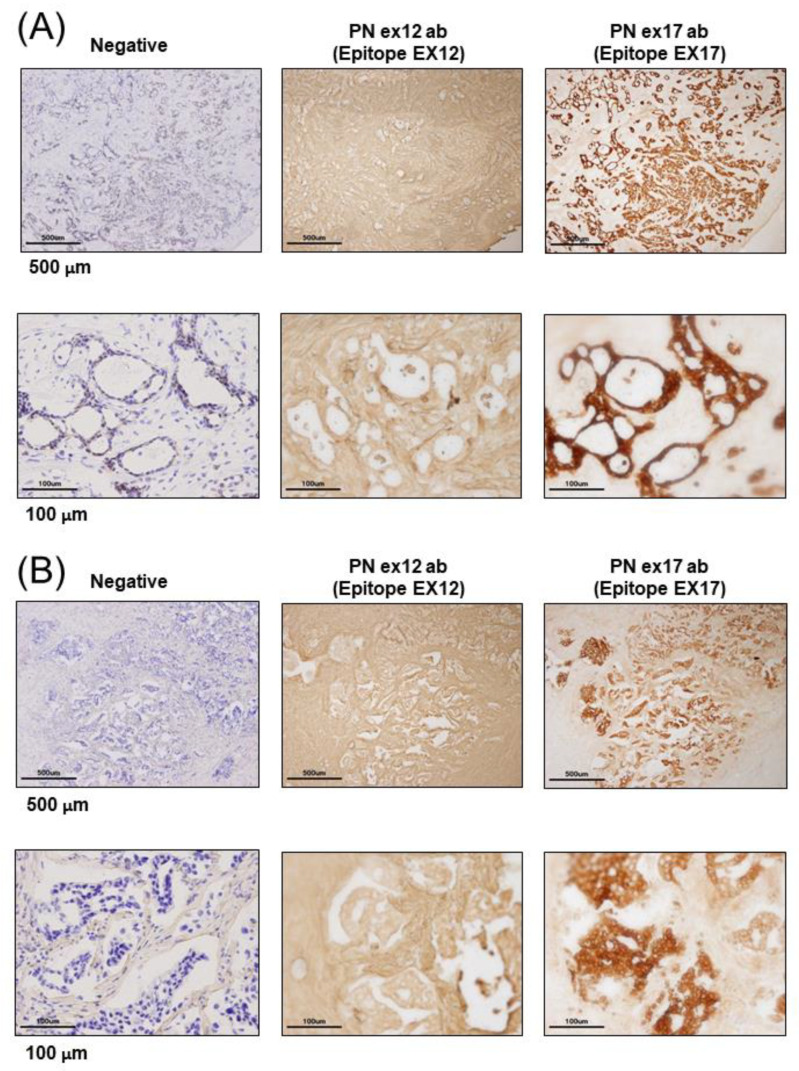
Expression of Periostin (POSTN) in human and mouse breast cancer. (**A**–**E**) Expression of POSTN in breast cancer patients. Tissue sections selected from two different breast cancer patients. The stained area represents positive staining for the POSTN antibody against each POSTN exon (original magnification, x4 in upper, and x20 in lower). Ex12 antibodypositive area is almost all stroma, and Ex17 antibody-positive area is almost all breast cancer cells. These areas are almost opposite. (**F**) Tissue sections with primary or metastatic lesions in the mouse triple-negative breast cancer (TNBC) xenograft model. We implanted 2 × 10^6^ cells of the MDA-MB-231 cell line in the mammary fat pads of 8-week-old female NOG mice. Eight weeks after implantation, we evaluated the primary and metastatic lesions. The upper section is primary lesion (Subcutaneous tissue of the mammary gland), and the lower section is metastatic lesion (lung). Ex12 antibody or Ex17 antibody positive areas are green and nuclei areas are blue. The Ex12 antibody-positive area is almost all stroma and the Ex17 antibody positive area is almost all breast cancer cells. These areas are almost all opposite. (**G**) POSTN exon and domain structure: the secreted protein, POSTN, has a signal sequence, one EMI, and four fasciclin 1 (FAS1) domains in the N-terminal region. The POSTN variants without exon 17, exon 21, or both in the C-terminal region, shown by alternative splicing. (**H**,**I**) Epitope mapping of POSTN antibodies. We synthesized POSTN human exon 1–23 peptides, and we applied each of them to 2–5 μL of 1 μg/mL on the nitrocellulose membrane separately. After this, we combined commercially available POSTN antibody or our Ex17 antibody [23,25]. Commercially available POSTN antibody (Ex12 antibody) detects POSTN Ex12 (H, yellow), and the Ex17 antibody detected POSTN Ex17 (I, yellow).

**Figure 2 cells-10-00892-f002:**
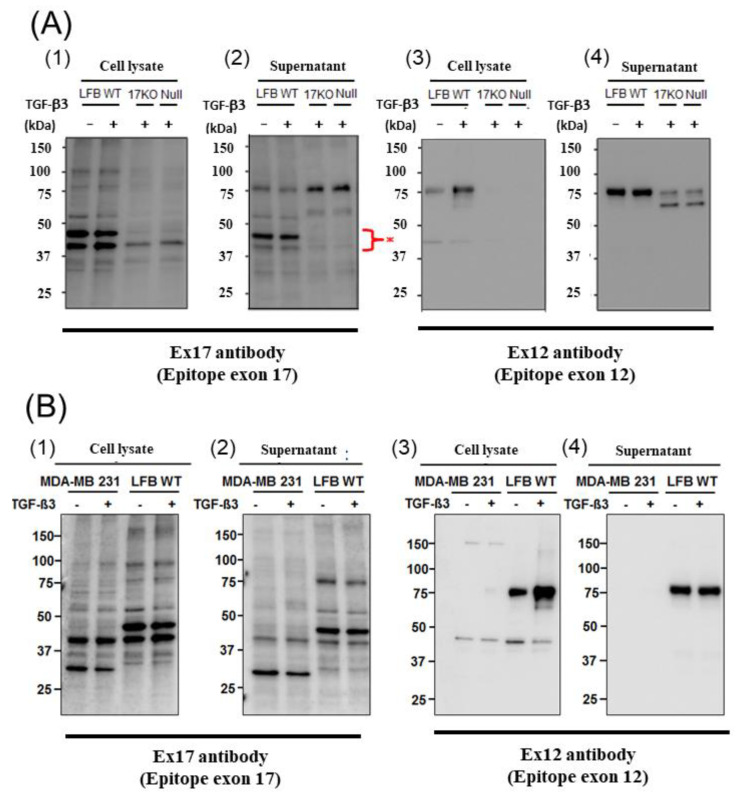
Western blotting of POSTN (**A**): The cell lysate (1) and supernatant (2) from lung fibroblasts (LFB) of wild-type (WT), 17KO (POSTN exon 17-absent mice) and null (POSTN-null mice) were applied after the immunoprecipitation of the Ex17 antibody. We performed Western blotting with the Ex17 antibody. We detected 2 bands between 50 and 37 kDa (*). The cell lysate (3) and supernatant (4) from lung fibroblasts (LFB) of WT, 17KO (POSTN exon 17-absent mice) and null (POSTN-null mice) were applied. We performed Western blotting with the Ex12 antibody. (**B**): The cell lysate (1) and supernatant (2) from MDA-MB-231 and lung fibroblasts (LFB) of WT with-or-without TGF-β after the immunoprecipitation of the Ex17 antibody. We performed Western blotting with the Ex17 antibody. We detected 2 bands between 50 and 37 kDa. The cell lysate (3) and supernatant (4) from MDA-MB-231 and lung fibroblasts (LFB) of WT with-or-without TGF-β. We performed Western blotting with the Ex12 antibody. We detected bands around 75 kDa.

**Figure 3 cells-10-00892-f003:**
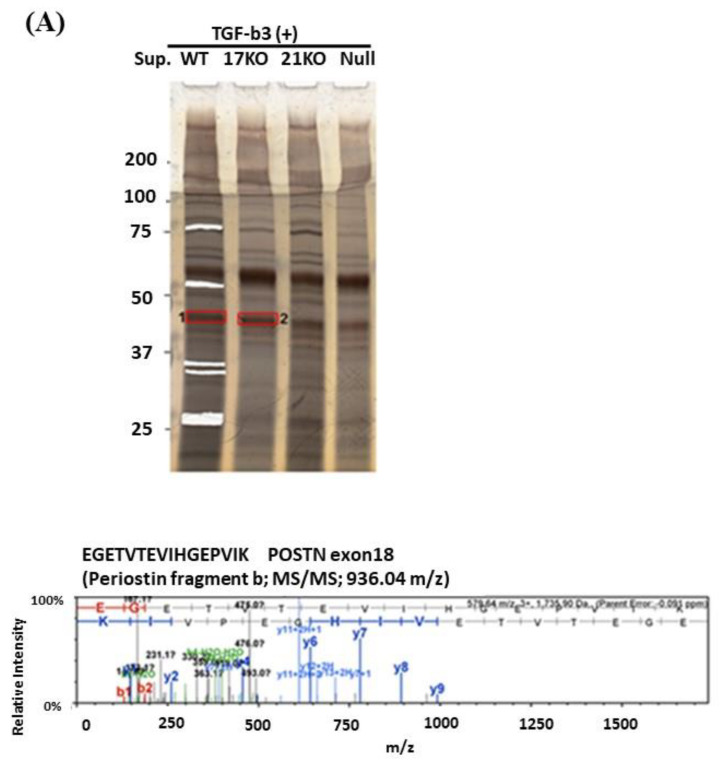
LC–MS/MS analysis. (**A**): Lung fibroblasts isolated from C57BL/6N WT, POSTN exon17KO, and POSTN-null mice were cultured in DMEM medium with 10% FBS and 1% penicillin/streptomycin, then the serum-free DMEM medium, including TGF-β, was replaced. After the stimulation of TGF-β, each supernatant was collected and concentrated using ultrafiltration and applied to SDS-PAGE after immunoprecipitation with the Ex17 antibody. The cut gel after silver stain was analyzed using LC–MS/MS. (**B**): MDA-MB-231 cells or lung fibroblasts of C57BL/6N WT mice were cultured in DMEM medium with 10% FBS and 1% penicillin/streptomycin, and the total protein from each cell lysate, 10, 30, and 100 µg, were applied for the SDS-PAGE. After Western blotting with Ex12 antibody, the 75 kDa position, surrounded by the black square, was analyzed with MALDI-QqTOF MS/MS.

**Figure 4 cells-10-00892-f004:**
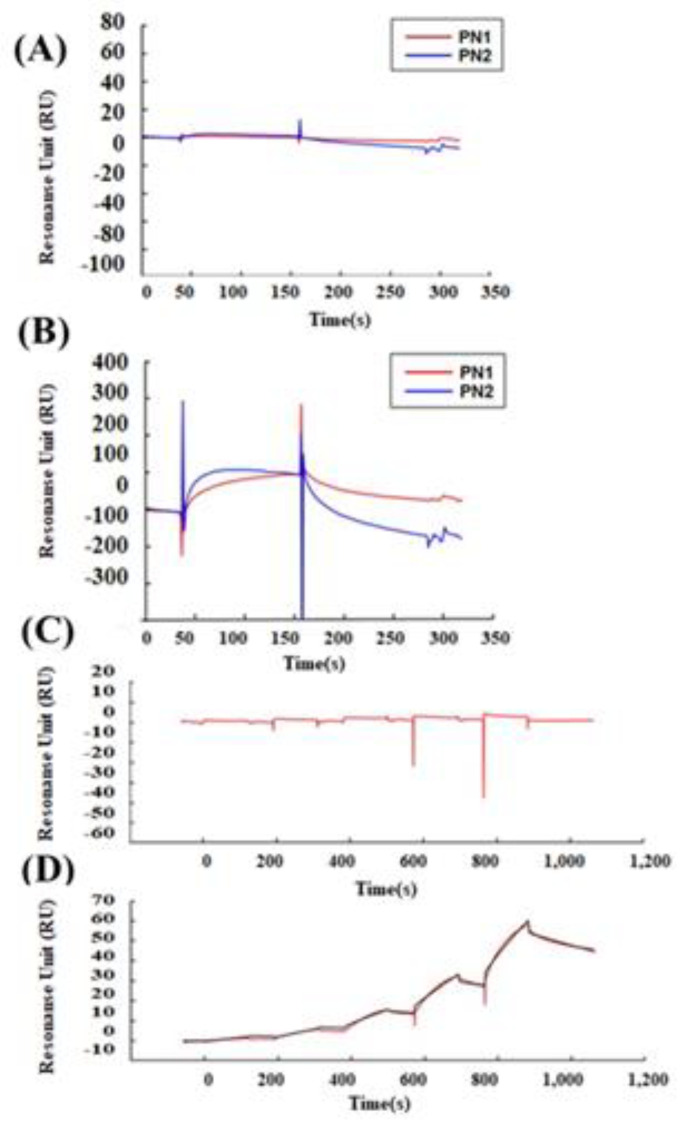
Protein–protein interaction between POSTN variants and wnt3a. Interaction between POSTN variants and wnt3a was estimated by surface plasmon resonance (SPR) analysis. A dilution series of BSA and wnt3a was prepared at five different concentrations of 0.062, 0.185, 0.556, 1.667, and 5 µM, and sequentially injected over a sensor surface prepared with each POSTN variant. After a contact time of 1 min, the flow was switched to the running buffer alone for another 2 min in a single cycle. The binding kinetic parameters (ka, kd, and KD) at each analyte concentration were measured, and one analysis was repeated twice to assess the accuracy of the data. Equilibrium dissociation constants (KD) are shown. Wnt3a (KD = 5.632 × 10^−9^). (**A**) SPR analysis between BSA and PN1 (red line) or PN2 (blue line). (**B**) SPR analysis between wnt3a and PN1 (red line) or PN2 (blue line). (**C**) Single-cycle kinetics approach between wnt3a and BSA (red line). (**D**,**E**) Single-cycle kinetics approach between wnt3a and PN1 (red line). Because wnt3a binds PN1 but not PN2 or BSA, we can only make the fitting curve between wnt3a and PN1. (**D**,**E**) shows the wnt3a dose-depending bindings to PN1. (**E**) The equations of the fitting curve were automatically created (black line) when the kinetic rate constants were extracted from the experimental data by an iterative process, and these fitting data are shown as a table. Ex17 antibody inhibited the binding between POSTN exon 17 and wnt3a. PN1 or a mixture of PN1 and wnt3a was injected after the POSTN exon 17 was immobilized on a CM5 sensor chip. (**F**) The binding between the Ex17 antibody and PN1 indicated by a red line; PN1 and BSA indicated by a blue line; Ex17 antibody and BSA indicated by a green line. BSA did not inhibit the binding of the Ex17 antibody and PN1. (**G**) The binding between the Ex17 antibody and PN1 is indicated by a red line; between PN1 and wnt3a indicated by a blue line; between Ex17 antibody and BSA indicated by a green line. The addition of wnt3a inhibited the binding of the Ex17 antibody and PN1 by 46.59%. The formula for the inhibition efficiency is shown below. [Resonance Unit (RU) of the mixture of PN1 and wnt3a]/[RU of the mixture of PN1] + [RU of the mixture of wnt3a] × 100 (%).

**Figure 5 cells-10-00892-f005:**
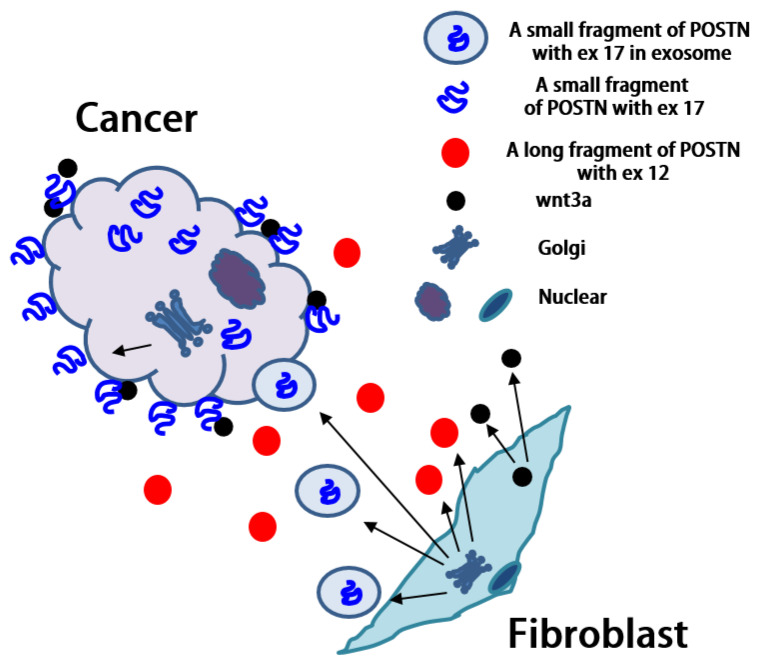
Possible mechanism of POSTN expression. Luga et al. reported that fibroblasts in the stroma promote breast cancer cell protrusive activity [32]. Fibroblasts may secrete exosomes (circle), including a short-fragment POSTN with Ex17 (blue) through Golgi modification, and the exosomes may move to cancer cells. On the other hand, fibroblasts also secrete POSTN without Ex17 (red), and it stays near fibroblasts in stroma. Ex17 antibody can detect the POSTN Ex17 protein in cancer cells but not in the stroma, as the Ex17 antibody cannot detect exosomes in the stroma. POSTN Ex12 antibody cannot detect a short fragment POSTN with Ex17 in the cancer area, as a short fragment would not have POSTN ex 12 considering with its molecular weight of about 40 kDa (whole POSTN weighs about 90 kDa). As a result, the Ex12 antibody can detect POSTN Ex12 in the stroma. In addition, wnt3a fibroblasts bind with short fragments of POSTN with Ex17 on the cancer cells.

## Data Availability

The data that support the findings of this study are available from the corresponding author, Y. Taniyama, upon reasonable request.

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
