# Peer review of "Periostin Short Fragment with Exon 17 via Aberrant Alternative Splicing Is Required for Breast Cancer Growth and Metastasis"

_cells, 2021, doi:10.3390/cells10040892_

Round 1
Reviewer 1 Report
This manuscript focuses on a matricellular protein Periostn (PN) which is predominantly synthesized and secreted by fibroblasts. This ECM protein has been shown to promote cancer metastasis, angiogenesis and the epithelial to mesenchymal transition as well as in inflammation and resistance to chemotherapy. Depending on the exons in splice variants, it is hypothesized that various splice variants may play distinct roles in cancer cell stromal interactions. Building on a previous study (Kyutoku, M. et al., 2011), suggesting that antibodies against exon 17 of PN C-terminal inhibited TNBC growth and metastasis to lungs, the authors propose that the C-terminal of PN might be important in breast cancer progression and metastasis. and investigated the mechanisms by which PN inhibits tumor growth. The authors demonstrate that the PN variant in the stroma is distinct from that in cancer cells based on the distinct detection by antibodies directed against the N-terminal Ex12 (stroma) and the C-terminal Ex17 (cancer cells). They also show that exon 17 containing PN interacted with wnt3a while PN lacking exon 17 did not. Finally, the study shows that exon 17 containing PN was mostly detected on the surface of TNBC cells. Overall, these data confirm the significance of exon 17 containing PN and suggest that this fragment of PN may be associated with the cell surface. While these findings are relevant in therapeutic targeting of breast cancer growth and metastasis, and therefore, improve our understanding of the putative C terminal fragment(s) of PN, the manuscript in its present form lacks consistency, poorly presented and requires extensive professional language editing.
Major issues;
- POSTN C-terminal antibody is that raised against Ex17, the subject of this manuscript. POSTN C-terminal antibody has already been shown to influence TNBC growth and metastasis to lungs (Kyutoku, M 2011). The animal studies in section 3.2 essentially duplicate previous studies by the same authors and should be excluded or discussed/presented only in the context of detection of PN-ex17 in the stroma versus cancer cells.
- The manuscript Figures should be presented chronologically and in the order they appear in the manuscript.
- Similarly, the figure legends should specifically address each figure in a systematic manner. g. the legend of Figure 1 begins with panel B and then Table 2...
- In Figure 1, it is not clear whether each panel (A and B) represent a single patient. It is also alleged that eight patient tissues were examined but the data appears to be for tissues from two patients.
- The rationale/background for the interaction of PN and wnt3a should be provided.
- The DuoLink assay (section 3.5; Figure 5B) should be complemented with surface biotinylation and streptavidin affinity pull down assay to demonstrate that the interaction is occurring on the cell surface.
- Figure 9, citing the secretion of exosomes by fibroblasts is speculative. This could be improved and the legend should include appropriate details including the relevant references.
Minor issues:
In the Abstract,
- The objective/hypothesis of the study should be clearly stated.
- The interaction between PN-40kDa fragment and wnt3a and the inhibition of the interaction by Ex17 abs is additional data from the study not the conclusion.
- Authors should provide a valid conclusion based on the data and the hypothesis/objective of the study.
There are several language and editing issues throughout the manuscripts. A few of these include:
Line 37, TNBC occurs in…10% of breast cancer patients. Since this deviates a little from the 10-15 % of breast cancers, please provide a reference to this statement.
Line 46, please revise the statement POSTN is a unique….insect cell adhesion
Lines 60-63, the hypothesis is difficult to understand. Please revise/restate.
Line 99, please clarify what is meant by clinical pathological organs (be more specific)
Line 295, Figure 7A seems to be out of place.
Author Response
Reviewer 1: Comments and Suggestions for Authors
This manuscript focuses on a matricellular protein Periostn (PN) which is predominantly synthesized and secreted by fibroblasts. This ECM protein has been shown to promote cancer metastasis, angiogenesis and the epithelial to mesenchymal transition as well as in inflammation and resistance to chemotherapy. Depending on the exons in splice variants, it is hypothesized that various splice variants may play distinct roles in cancer cell stromal interactions. Building on a previous study (Kyutoku, M. et al., 2011), suggesting that antibodies against exon 17 of PN C-terminal inhibited TNBC growth and metastasis to lungs, the authors propose that the C-terminal of PN might be important in breast cancer progression and metastasis. and investigated the mechanisms by which PN inhibits tumor growth. The authors demonstrate that the PN variant in the stroma is distinct from that in cancer cells based on the distinct detection by antibodies directed against the N-terminal Ex12 (stroma) and the C-terminal Ex17 (cancer cells). They also show that exon 17 containing PN interacted with wnt3a while PN lacking exon 17 did not. Finally, the study shows that exon 17 containing PN was mostly detected on the surface of TNBC cells. Overall, these data confirm the significance of exon 17 containing PN and suggest that this fragment of PN may be associated with the cell surface. While these findings are relevant in therapeutic targeting of breast cancer growth and metastasis, and therefore, improve our understanding of the putative C terminal fragment(s) of PN, the manuscript in its present form lacks consistency, poorly presented and requires extensive professional language editing.
We appreciate your comments and criticisms of our manuscript. We are very happy to hear that “These findings are relevant in therapeutic targeting of breast cancer growth and metastasis”. We have improved our manuscript in compliance with your suggestions.
Major issues;
Q1. POSTN C-terminal antibody is that raised against Ex17, the subject of this manuscript. POSTN C-terminal antibody has already been shown to influence TNBC growth and metastasis to lungs (Kyutoku, M 2011). The animal studies in section 3.2 essentially duplicate previous studies by the same authors and should be excluded or discussed/presented only in the context of detection of PN-ex17 in the stroma versus cancer cells.
A1. Previous paper, we demonstrated that rabbits polyclonal antibody against PN ex17 inhibits the primary and metastatic growth of mice TNBC (Kyutoku, M 2011), and in this paper, we demonstrated that mice monoclonal antibody for PNex17 inhibits the primary and metastatic growth of human TNBC. The antibody and cancer cell are different, but the meaning may be same. According to your suggestion, we excluded this animal study (Fig. 2 and 7).
- The manuscript Figures should be presented chronologically and in the order they appear in the manuscript. Similarly, the figure legends should specifically address each figure in a systematic manner. g. the legend of Figure 1 begins with panel B and then Table 2.
A2. Thank you for the suggestion, we made a word file with figures chronologically, but it happen to change. I don’t know the reason, but we will consult with the editor of Cells.
- In Figure 1, it is not clear whether each panel (A and B) represent a single patient. It is also alleged that eight patient tissues were examined but the data appears to be for tissues from two patients.
A3. Thank you for the suggestion, we added “Tissue sections from breast cancer patients, which were selected different 2 patients (A) and (B).”, and changed “when we examined the 2 eight breast cancer patients” in Result 3.1.
- The rationale/background for the interaction of PN and wnt3a should be provided.
A4. Thank you for the suggestion, we added “Wnta3a was reported to bind to chondroitin sulfate proteoglycans (one of the matrix proteins) existing on the cell surface [37]. POSTN, a matrix protein, also exists in the proteoglycan layer on the cell surface. The combination between wnt3a and POSTN appears natural” in Discussion.
- The DuoLink assay (section 3.5; Figure 5B) should be complemented with surface biotinylation and streptavidin affinity pull down assay to demonstrate that the interaction is occurring on the cell surface.
A5. Malanchi et al. showed the interaction between SBP/His-tagged POSTN and HA-tagged wnt3a transfected into 293T cells by pull down and performed western blotting.
We tried surface biotinylation and streptavidin affinity pull down assay, and failed. It was so difficult for us to detect the endogenous wnt3a protein in cultured cancer cell line MDA-MBA-231 cells even if we used LC-MS/MS. We thought that the POSTN exon 17 expression on cancer cell surface in vitro is not so high. When we tried to detect POSTN and wnt3a after cell surface biotinylation of MDA-MBA-231 or lung fibroblast, only POSTN with exon 17 could be detected from the cell surface fraction. Because the expression level of the endogenous proteins was not enough, we changed to use the surface plasmon resonance analysis to confirm the binding between biotinylated POSTN and wnt3a.
- Figure 9, citing the secretion of exosomes by fibroblasts is speculative. This could be improved and the legend should include appropriate details including the relevant references.
A6. Thank you for the suggestion, we sited Luga et al. 2012, Cell: Exosomes mediate stromal mobilization of autocrine Wnt-PCP signaling in breast cancer cell migration, and added “Luga et al. reported that fibroblasts in the stroma promote breast cancer cell protrusive activity via wnt signaling [32]”in Fig. 7 legend.
Minor issues:
In the Abstract,
- The objective/hypothesis of the study should be clearly stated.
A7. According to your suggestion, we added “Objective: The aim of this study is to investigate the POSTN function on the cancer.” in Abstract.
- The interaction between PN-40kDa fragment and wnt3a and the inhibition of the interaction by Ex17 abs is additional data from the study not the conclusion.
A8. Thank you for the suggestion, we added “Proximity ligation assay also showed that Ex17 Abs inhibits the same binding on the cancer cell surface.” in Abstract. DuoLink assay is a product name, so we changed it to proximity ligation assay
- Authors should provide a valid conclusion based on the data and the hypothesis/objective of the study.
A9. Thank you for the suggestion, we added Objective and a valid conclusion in Abstract. “Objective: This study aimed to investigate the function of POSTN on tumors” and “proximity ligation assay showed that the Ex17 antibody inhibited the same binding on the cancer cell surface.”
Q10: There are several language and editing issues throughout the manuscripts. A few of these include: Line 37, TNBC occurs in…10% of breast cancer patients. Since this deviates a little from the 10-15 % of breast cancers, please provide a reference to this statement.
A10: Thank you for the suggestion, we cited a new reference as follow. Dass, S. A. et al. 2021, Medicina (Kaunas) :Triple Negative Breast Cancer: A Review of Present and Future Diagnostic Modalities.
Q11. Line 46, please revise the statement POSTN is a unique….insect cell adhesion
A11. Thank you for the suggestion, we changed sentences as follow. POSTN is a unique protein that it shares homology with the insect cell adhesion molecule fasciclin I (FASI) or the human IgH3 (TGFBI) protein, both of them which are induced by transforming growth factor- (TGF- )
Q12. Lines 60-63, the hypothesis is difficult to understand. Please revise/restate.
A12. Thank you for the suggestion, we revised the sentences as follows. We hypothesized that the localization of POSTN fragment could vary in cancer-stromal interactions if it is assumed that there is a POSTN fragment protein with exon 17 corresponding to the number of exons in the POSTN C-terminus region but not exon 12 in the POSTN N-terminus region.
Q13. Line 99, please clarify what is meant by clinical pathological organs (be more specific)
A13: Thank you for the suggestion, we changed sentences as follow. “but the function of POSTN exon 17 in clinical human pathological organs such as primary and metastatic region of breast cancer is still unclear.”
Q14: Line 295, Figure 7A seems to be out of place.
A14. Thank you for the suggestion, but we excluded the result of animal study (Fig.7).
Once again we appreciate your criticisms and we do hope that our revision will meet with your approval and that you will now find our paper acceptable for publication in Cells.
Reviewer 2 Report
The publication "Periostin short fragment including exon 17 via aberrant alternative splicing is required for breast cancer growth and metastasis" is an interesting scientific study based on in vivo (mice) and in vitro (6 cell cultures) models. The results of the publication have a potential clinical aspect.
The publication is prepared in a concise and understandable way.
The only reservations are raised by the numbering of figures - figure 6 is first, then 1, 7, 2, 3, 8, 4, 5. The Authors should arrange it.
The size of all figures should be increased as they are illegible in their current form.
The bibliography items have been published in the last 15-20 years and are properly selected.
I think the work is topic spotty and should be published in the Cells.
Author Response
Reviewer 2: Comments and Suggestions for Authors
The publication "Periostin short fragment including exon 17 via aberrant alternative splicing is required for breast cancer growth and metastasis" is an interesting scientific study based on in vivo (mice) and in vitro (6 cell cultures) models. The results of the publication have a potential clinical aspect. The publication is prepared in a concise and understandable way. The only reservations are raised by the numbering of figures - figure 6 is first, then 1, 7, 2, 3, 8, 4, 5. The Authors should arrange it. The size of all figures should be increased as they are illegible in their current form. The bibliography items have been published in the last 15-20 years and are properly selected.
I think the work is topic spotty and should be published in the Cells.
We appreciate your comments and criticisms of our manuscript. We are very happy to hear that “I think the work is topic spotty and should be published in the Cells.” We have improved our manuscript in compliance with your suggestions. We arranged the Figures order, and increase the size of figures. Once again we appreciate your criticisms and we do hope that our revision will meet with your approval and that you will now find our paper acceptable for publication in Cells.
Reviewer 3 Report
Title: Periostin short fragment including exon 17 via aberrant alterna-2 tive splicing is required for breast cancer growth and metastasis
Authors: Yuka Ikeda-Iwabu, Yoshiaki Taniyama, Naruto Katsuragi, Fumihiro Sanada, Nobutaka Koibuchi , Kana Shi-bata, Kenzo Shimazu, Hiromi Rakugi and Ryuichi Morishita.
In this manuscript, the authors looking into the possible role that the POSTN C-terminus may play in breast cancer progression and metastasis. The authors compared the distribution of POSTN variants in breast cancer using antibodies against POSTN exon 17 (C-terminal) and POSTN exon 12 (N-terminal), and the function of POSTN exon 17 in vivo and in an in vitro using the triple negative breast cancer animal model.
Concerns:
- The manuscript requires major language editing. There are several grammatical errors throughout the manuscript, which interfere with understanding the author's intentions.
- The figure legends do not match with the figures and are very confusing. In Figure 1, the authors refer to table 2, although the legend should be explaining the referred figure. The sentence starts with the letter “B” followed by a second sentence saying “patients (A) and (B) which has no exact meaning.
- Reorientation of the figure is needed to have a logical flow that fits the text in the manuscript. The authors discuss figure 1, followed by figure 7, and figure 3, followed by figure 8. The order of the figure numbers referenced throughout the manuscript should follow the order of the figures.
- In the results section 3.1 line 273-275, the authors mentioned that “Conversely, the sections from 273 breast cancer patients were completely stained in the stromal and cancer cells slightly 274 stained when a commercially available monoclonal antibody against exon 12 of POSTN 275 was used (Fig. 1A-B) of which we checked the epitope”. It is not clear what the authors mean by we checked the epitope; The authors need to provide a clear explanation.
- In the result section, line 294-296, the authors mentioned that they looked at the subcutaneous primary tumor's immunohistochemistry and the metastatic site without clearly defining what we are looking at, what is the green stain versus blue. No explanation in the figure legend; please add a more detailed explanation of figure 1C.
- In figure 2, B, the authors mention that the Ex17 Ab treatment reduced the primary tumor volume, but the difference seems very trivial. In the methods section, the authors stated that “treatment started once the primary tumor was established, approximately eight weeks after cell implantation,” while the figures show the Ab treatment effect after eight weeks. That means the effect of the Ab treatment detected the time it was administered; please explain.
Author Response
Reviewer 3: Comments for the Author
In this manuscript, the authors looking into the possible role that the POSTN C-terminus may play in breast cancer progression and metastasis. The authors compared the distribution of POSTN variants in breast cancer using antibodies :13against POSTN exon 17 (C-terminal) and POSTN exon 12 (N-terminal), and the function of POSTN exon 17 in vivo and in an in vitro using the triple negative breast cancer animal model.
We appreciate your comments and criticisms of our manuscript. We have improved our manuscript in compliance with your suggestions.
Concerns:
Q1. The manuscript requires major language editing. There are several grammatical errors throughout the manuscript, which interfere with understanding the author's intentions.
A1. Thank you for the suggestion, we again did language editing.
Q2. The figure legends do not match with the figures and are very confusing. In Figure 1, the authors refer to table 2, although the legend should be explaining the referred figure. The sentence starts with the letter “B” followed by a second sentence saying “patients (A) and (B) which has no exact meaning.
A2 Thank you for the suggestion, we
Q3.Reorientation of the figure is needed to have a logical flow that fits the text in the manuscript. The authors discuss figure 1, followed by figure 7, and figure 3, followed by figure 8. The order of the figure numbers referenced throughout the manuscript should follow the order of the figures.
A3. Thank you for the suggestion, we made a word file with figures chronologically, but it happen to change. I don’t know the reason, but we will consult with the editor of Cells.
Q4. In the results section 3.1 line 273-275, the authors mentioned that “Conversely, the sections from 273 breast cancer patients were completely stained in the stromal and cancer cells slightly 274 stained when a commercially available monoclonal antibody against exon 12 of POSTN 275 was used (Fig. 1A-B) of which we checked the epitope”. It is not clear what the authors mean by we checked the epitope; The authors need to provide a clear explanation.
A4. Thank you for the suggestion, we change the sentences to “Conversely, the sections from breast cancer patients were completely stained in the stromal and cancer cells slightly stained using a commercially available POSTN monoclonal antibody (Ex12 Abs) (Fig. 1A-B) We confirmed its epitope POSTN exon 12 by dot blotting. (Fig. 6)” in Result.
Q5. In the result section, line 294-296, the authors mentioned that they looked at the subcutaneous primary tumor's immunohistochemistry and the metastatic site without clearly defining what we are looking at, what is the green stain versus blue. No explanation in the figure legend; please add a more detailed explanation of figure 1C.
A5. Thank you for the suggestion, we added “Ex12 Abs or Ex17 antibody positive area was green and nuclei area was blue. Ex12” in figure 1C legend.
Q6. In figure 2, B, the authors mention that the Ex17 Ab treatment reduced the primary tumor volume, but the difference seems very trivial. In the methods section, the authors stated that “treatment started once the primary tumor was established, approximately eight weeks after cell implantation,” while the figures show the Ab treatment effect after eight weeks. That means the effect of the Ab treatment detected the time it was administered; please explain.
A6. Thank you for the suggestion. Because the size of each cancer is very different, we administered Ex17 Abs after 8 weeks. Your understanding is right, but reviewer 1 recommend that the animal studies in section 3.2 essentially duplicate previous studies by the same authors and should be excluded.
Previous paper, we demonstrated that rabbits polyclonal antibody against PN ex17 inhibits the primary and metastatic growth of mice TNBC (Kyutoku, M 2011), and in this paper, we demonstrated that mice monoclonal antibody for PNex17 inhibits the primary and metastatic growth of human TNBC. The antibody and cancer cell are different, but the meaning may be same. According to reviewer 1’s suggestion, we excluded this animal study.
Once again we appreciate your criticisms and we do hope that our revision will meet with your approval and that you will now find our paper acceptable for publication in Cells.

Round 2
Reviewer 1 Report
Revised version:
I this revised version, the authors have attempted to restructure the manuscript accordingly. In spite of the apparent improvement, the description of experiments and experimental data remains below par. Also the novelty of the findings is hard to discern as the main objective of demonstrating the interaction of POSTN with Wnt3a is not adequately demonstrated and it remains unclear which of the POSTN isoforms is expressed in breast cancer cells. Furthermore, several issues either remain unresolved or need to be effectively resolved. These include:
- The confirmation of the interaction between POSTN and Wnt3a on the cell surface needs to be carried out by other assays. Surface Plasmon Resonance (Biacore) analysis does not suggest the interaction is occurring on the cell surface. If the interaction cannot be confirmed on the cell surface, then this should not be reported; and given that the interaction between these proteins is well established, the data is simply confirming known facts.
- The notion that fibroblast derived exosomes can promote breast cancer cell protrusive activity via Wnt signaling [Ref 32] does not suggest a role for POSTN/Wnt3a in this cell behavior. Additional data including detection of POSTN in EVs as well as the effects of POSTN containing and POSTN devoid EVs on cancer cell protrusive activity need to be provided.
- The PLA assay as described does not also suggest that the interaction of the proteins occurred on the cell surface given that methanol fixation allows for antibody infusion into cells.
- Ex-17 is present in PN1 and PN3 but not PN2 and PN4. It is not clear whether detection of POSTN in breast cancer tissues with Ex17 Ab suggest either of these variants or new variants of POSTN? It is also not clear whether the 40 kDa fragment originates from one or more of the Ex17 containing POSTN. The study could be more focused by addressing which of these Ex17 containing POSTNs is more relevant in breast cancer.
- It is still unclear why the authors are addressing Ex 17 containing POSTN as Ex17 POSTN. Results from section 3.1 suggest that breast cancer cells express Ex17 containing POSTNs while the stroma expressed mostly Ex12 containing POSTN isoforms. If this has not been demonstrated elsewhere, then it will be better to clarify this by detection of the various POSTN isoforms in breast cancer cells.
- Fig. 1 is still poorly interpreted given that tissues from only two breast cancer patients were analyzed. The very strong conclusions does not adequately represent the heterogeneous nature of breast cancer. It will be nice to include the specific breast cancer subtype and more importantly, quantitative analysis of several tissues with these antibodies.
- Fig. 3E appears to describe the kinetic parameters for the binding of Wnt3a to PN1, PN2 or BSA. It is not clear what interaction/interacting partner the kinetic parameters refer to.
- The experiments described in section 3.4 (Fig. 4A) do not suggest that the well-established interaction between POSTN and Wnt3a occurs on the cell surface. The marker used in this case WAGA is not specific for cell membranes and the methanol fixation of cells permeabilizes the cells allowing detection of intracellular and presumably cell surface glycoproteins.
- Similarly, the use of Ex17 antibody to inhibit the interaction of POSTN and Wnt3a by PLA should be clearly described. The fact that cells are exposed to the purified proteins prior to PLA makes this procedure even more intriguing. However, the authors should indicate the primary antibodies used for the PLA and whether the POSTN antibody is different from EX17 antibody. GST pull-down assays with or without the EX17 antibody should provide more reliable data.
Other comments:
- The manuscript is still very poorly written with extremely difficult descriptions of experiments and especially Figure legends.
- The description of the PLA procedure is unclear, incoherent and confusing for a well-established protocol.
- Fig. 1E and F; Please indicate the exon numbers on the dot blot to clarify.
- Line 366. Figures are not still indicated in the manuscript in a chronological manner. Here from Fig. 2 to Fig. 5B etc.
- Fig. 5. Better images of the LC-MS/MS spectra should be presented.
Author Response
Attached please find the response to reviewer 1.

Reviewer 3 Report
The authors addressed all concerns adequetly
Author Response
Thank you so much for your positive comments.